# Zooplankton grazing of microplastic can accelerate global loss of ocean oxygen

K. Kvale [1,3 ✉], A. E. F. Prowe[1], C.-T. Chien[1], A. Landolfi [1,2] & A. Oschlies [1]

Global warming has driven a loss of dissolved oxygen in the ocean in recent decades. We demonstrate the potential for an additional anthropogenic driver of deoxygenation, in which zooplankton consumption of microplastic reduces the grazing on primary producers. In regions where primary production is not limited by macronutrient availability, the reduction of grazing pressure on primary producers causes export production to increase. Consequently, organic particle remineralisation in these regions increases. Employing a comprehensive Earth system model of intermediate complexity, we estimate this additional remineralisation could decrease water column oxygen inventory by as much as 10% in the North Pacific and accelerate global oxygen inventory loss by an extra 0.2–0.5% relative to 1960 values by the year 2020. Although significant uncertainty accompanies these estimates, the potential for physical pollution to have a globally significant biogeochemical signal that exacerbates the consequences of climate warming is a novel feedback not yet considered in climate research.

[1] GEOMAR Helmholtz Centre for Ocean Research, West shore campus, Kiel, Germany. [2] ISMAR-CNR, Roma, Italy. [3] Present address: GNS Science, Lower Hutt, New Zealand. ✉email: kkvale@geomar.de

The global ocean is losing oxygen[1]. Drivers of this loss have been ascribed to climate change and the associated warming and altered circulation, as well as indirect effects on biogeochemistry[2]. However, climate change is not the only stressor on ocean biology, and therefore biogeochemistry. Plastic pollution in the global ocean is also increasing[3]. A growing body of work shows zooplankton consume the smaller size fractions of plastic[4,5]. These small particles (the microplastics, typically defined as having a length between 0.1 μm and 5 mm) replace food in the zooplankton's diet and reduce their consumption (and subsequent export) of particle-bound organic carbon[6]. If sufficiently widespread, this reduction of grazing on primary producers might have global biogeochemical consequences. Explicit consumption of microplastic by zooplankton, as well as microplastic aggregation in marine snow, was recently implemented in an Earth system model in order to simulate the transport of microplastic particles by biology[7]. They demonstrated a potentially significant influence of both marine snow (aggregated organic detrital material) and zooplankton faecal pellets in shaping microplastic distributions in the global ocean. Here we report that the zooplankton ingestion of microplastic in those same simulations also affects ocean biological rates relevant to dissolved oxygen. This result demonstrates that a physical effect of plastic pollution might presently have a disruptive influence on global ocean oxygenation equivalent of up to half that of climate warming, and it suggests a missing mechanism in Earth system models, which typically underestimate 21st century ocean deoxygenation[2].

## Results

**Inter-model parameterisation differences.** Biological uptake has the potential to profoundly shape microplastic particle distributions in the global ocean[7], concentrating particles in biologically-active surface regions as well as in gyres, and transporting particles to the deep ocean. Figure 1 displays simulated microplastic concentrations from a model that does not include biological uptake (No Bio) and from three models that do; 'Low Concentration' (LC), 'High Concentration' (HC), and 'Moderate Concentration' (MC). These three simulations are highlighted to represent the solution space of the 14 individuals of a 300 member perturbed parameter ensemble[7,8] that produced plausible global microplastic inventories[9] and subsurface particle maxima[10], using pollution rates[11] and marine snow aggregation rates[12] within available estimates. Microplastic model parameter values for each simulation presented here are provided in Table 1. Zooplankton relative grazing selectivities for microplastic vary between simulations; in the High Concentration simulation, zooplankton prefer other food sources and consequently have a low relative grazing selectivity for microplastic (the relative grazing selectivity parameter, $\psi_{MP} = 0.132$, with relative grazing selectivity for non-nitrogen fixing phytoplankton evenly divided over the remainder against 1[7]). The Moderate Concentration simulation uses a relative grazing selectivity for microplastic approximately balanced with other food sources ($\psi_{MP} = 0.193$), and the Low Concentration simulation uses a high relative grazing selectivity that favours microplastic consumption ($\psi_{MP} = 0.260$). The Low Concentration simulation also represents relatively enhanced marine snow/microplastic aggregation (which helps to explain the relatively lower surface microplastic concentrations in this simulation), while High Concentration simulation represents relatively reduced marine snow/microplastic aggregation. The Moderate Concentration simulation represents moderate marine snow/microplastic aggregation rates and contains an upper-ocean microplastic inventory roughly closest of the three to an observationally-constrained estimate of 'large' microplastics[9], which falls somewhere between the Moderate Concentration simulation and the High Concentration simulation. Model surface grid (to 50 m depth) total microplastic concentrations in the Moderate and High Concentration simulations are to the same order as a recently aggregated near-surface microplastic concentration dataset[13], although all parameter combinations fail to produce the very high concentrations recorded in gyres. This is not unexpected[7] due to the coarse resolution of our model. Whether microplastic concentrations are truly higher in subtropical gyres remains unclear[14]. The Low Concentration simulation

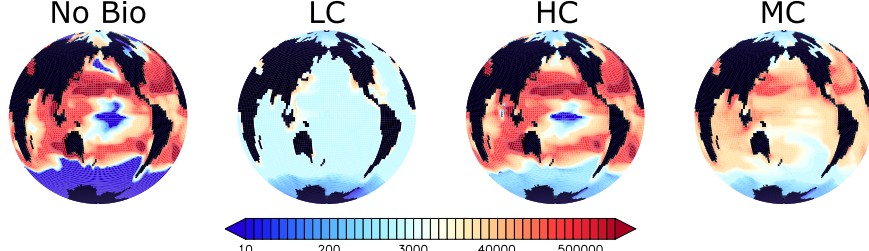

**Fig. 1 Near-surface depth-integrated total microplastic particle inventories at year 2020 in four simulations.** Abbreviations are LC (Low Concentration), HC (High Concentration), and MC (Moderate Concentration). Microplastic parameters are different between models and therefore produce different surface concentrations and particle inventories. Greater biological interaction with microplastic (e.g., larger marine snow/microplastic aggregation rates and larger zooplankton relative grazing selectivities for microplastic) result in lower near-surface microplastic inventories. Units are microplastic particles km$^{-2}$ in the top 100 meters of the ocean. Note the colour scale is logarithmic.

**Table 1 Parameter values used in the presented simulations.**

| Simulation | Fraction of total annual waste as microplastic | Fraction assigned a rise rate | Fraction returning to water column from seafloor | Marine snow aggregation fraction | Aggregate uptake of microplastic constant | Relative grazing selectivity | Food-to-microplastic conversion ratio |
|---|---|---|---|---|---|---|---|
| No Bio | 0.260 | 0.033 | – | – | – | – | – |
| LH | 0.137 | 0.003 | 0.132 | 0.003 | 0.424 | 0.260 | 1.029 |
| HC | 0.329 | 0.074 | 0.888 | 0.098 | 833.290 | 0.132 | 1.489 |
| MC | 0.276 | 0.011 | 0.528 | 0.092 | 615.508 | 0.193 | 0.993 |

A full model description can be found in our accompanying manuscript[7].

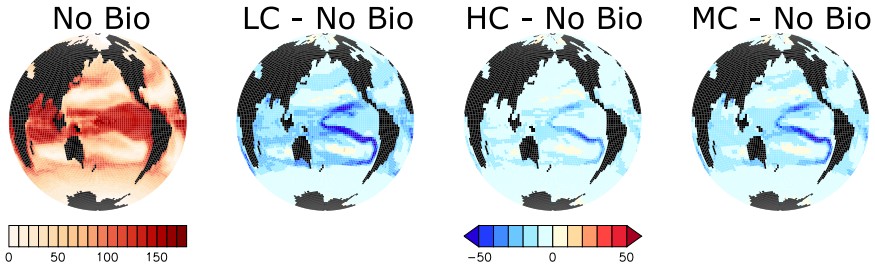

**Fig. 2 Simulated annual grazing flux of organic matter normalized by prey biomass (per year) at year 2020.** Differences between each of the three simulations that include biological uptake of microplastic and the No Bio simulation are shown together with absolute values for No Bio. Abbreviations are LC (Low Concentration), HC (High Concentration), and MC (Moderate Concentration).

does not simulate strongly increasing microplastic concentrations in gyres. A parameter sensitivity analysis[7], (their Supplementary Information) shows that aggregation of microplastic in marine snow exerts a strong control on microplastic distributions in this model, although microplastic storage in zooplankton faecal pellets might be the larger particle sink.

**Ecosystem response to zooplankton grazing of microplastic.** Microplastic represents for zooplankton a nutrition-less by-product being ingested alongside other food sources, that mostly reduces zooplankton consumption of primary producers and their associated nutrients (Fig. 2). Reduced consumption of primary producers alters the marine ecosystem in two directions, depending on whether or not the system is macronutrient-limited (Fig. 3): in regions where surface macronutrients are abundant, a reduction of grazing pressure on primary producers permits additional export production by phytoplankton. The reduction of grazing on phytoplankton conveys a greater portion of phytoplankton biomass directly into sinking detritus, which upon remineralisation consumes oxygen and returns nutrients at depth. However, in warm, oligotrophic subtropical regions where surface macronutrients are scarce and primary producers are reliant on regenerated nutrients, a reduction of grazing pressure cannot stimulate significant additional export production. Here, with warmer temperatures, the lack of grazing on phytoplankton conveys a greater portion of phytoplankton biomass into the microbial loop. Nutrients are trapped at the surface, leading to a decline in export production. Export production in macronutrient-limited regions decreases with zooplankton ingestion of microplastic as a consequence of both the reduction of nutrient regeneration via the excretion pathway as well as the relative enhancement of microbial loop recycling over zooplankton excretion for the return of nutrients to the surface ocean.

**North Pacific response.** Export production increases in response to zooplankton consumption of microplastic in the macronutrient-replete western North Pacific between 10–30% by year 2020 (Fig. 4). Higher rates of particle export result in more particles remineralising, which leads to additional consumption of water column dissolved oxygen (Fig. 5). The Low Concentration simulation demonstrates the largest difference in export production and oxygen inventory relative to the No Bio control simulation, while having the smallest microplastic surface concentrations (which might be expected to reduce zooplankton exposure to microplastic, but which also indicates efficient microplastic uptake by zooplankton). Likewise, the High Concentration simulation demonstrates the smallest export production and oxygen inventory differences, while having the largest microplastic surface concentrations. This can be explained by the zooplankton relative grazing selectivity for microplastic, which is the largest in the Low Concentration simulation and the smallest

in the High Concentration simulation. There are downstream consequences to this regional increase in export production; fewer nutrients are transported eastward, which results in net declines in export production in the eastern North Pacific.

**North Atlantic response.** A similar pattern is observable in the macronutrient-replete North Atlantic, although the magnitude is smaller (a 0–20% increase in export production by year 2020). As in the North Pacific, zooplankton grazing of microplastic results in enhanced new production, but because surface ocean temperatures are warmer, the microbial loop and detrital remineralisation rates are also relatively enhanced, which mitigates the increase in export production relative to the effect seen in the North Pacific. The consequences for water column oxygen are likewise mitigated (<10 mol $O_2$ m$^{-2}$ reduction in inventory by 2020; Fig. 5), in part because anomalies due to microbial processes occurring near the surface are masked by air-sea gas exchange. However, deoxygenation also has overall less biogeochemical relevance in this region due to the relatively high background oxygen levels in the well-ventilated North Atlantic.

**Tropical response.** Likewise, in the macronutrient-replete eastern tropical Pacific, grazing pressure on organic particles is reduced in response to microplastic contamination (Fig. 2). This leads to an increase in new production and particle export (of 0–30% depending on region and model configuration, Fig. 4), but not significant additional water column oxygen loss, because remineralisation occurs within suboxic zones, where nitrogen, rather than oxygen, is consumed. As in the North Pacific, there are downstream effects: increased local export production reduces westward nutrient transport out of the upwelling system. Particularly affected is nitrate, which is removed at an accelerated rate by denitrification in the eastern tropical Pacific. The resulting surface relative nitrate deficiency (Fig. 6) helps to suppress export production across the macronutrient-limited western tropical Pacific in all simulations in which microplastic is consumed by zooplankton (Fig. 3). In addition, the loss of zooplankton excretion as well as the relative enhancement of the microbial loop with respect to zooplankton excretion (Fig. 7) drives the reduction of export production by 10–40% across both the western tropical Pacific and tropical Atlantic. The reduction of export production leads to an increase in water column oxygen inventory by as much as 10 mol $O_2$ m$^{-2}$ by 2020 in both regions (Fig. 5), relative to the No Bio simulation.

Total export production decreases in the northern Indian Ocean by as much as 30% with the application of zooplankton grazing of microplastic (Fig. 4). In our model this region experiences seasonal shifts into macronutrient limitation not visible in the annually averaged map, with the northern Bay of Bengal being a (barely) nutrient-limited suboxic upwelling zone but the rest of the Bay, and the Arabian Sea, being macronutrient-

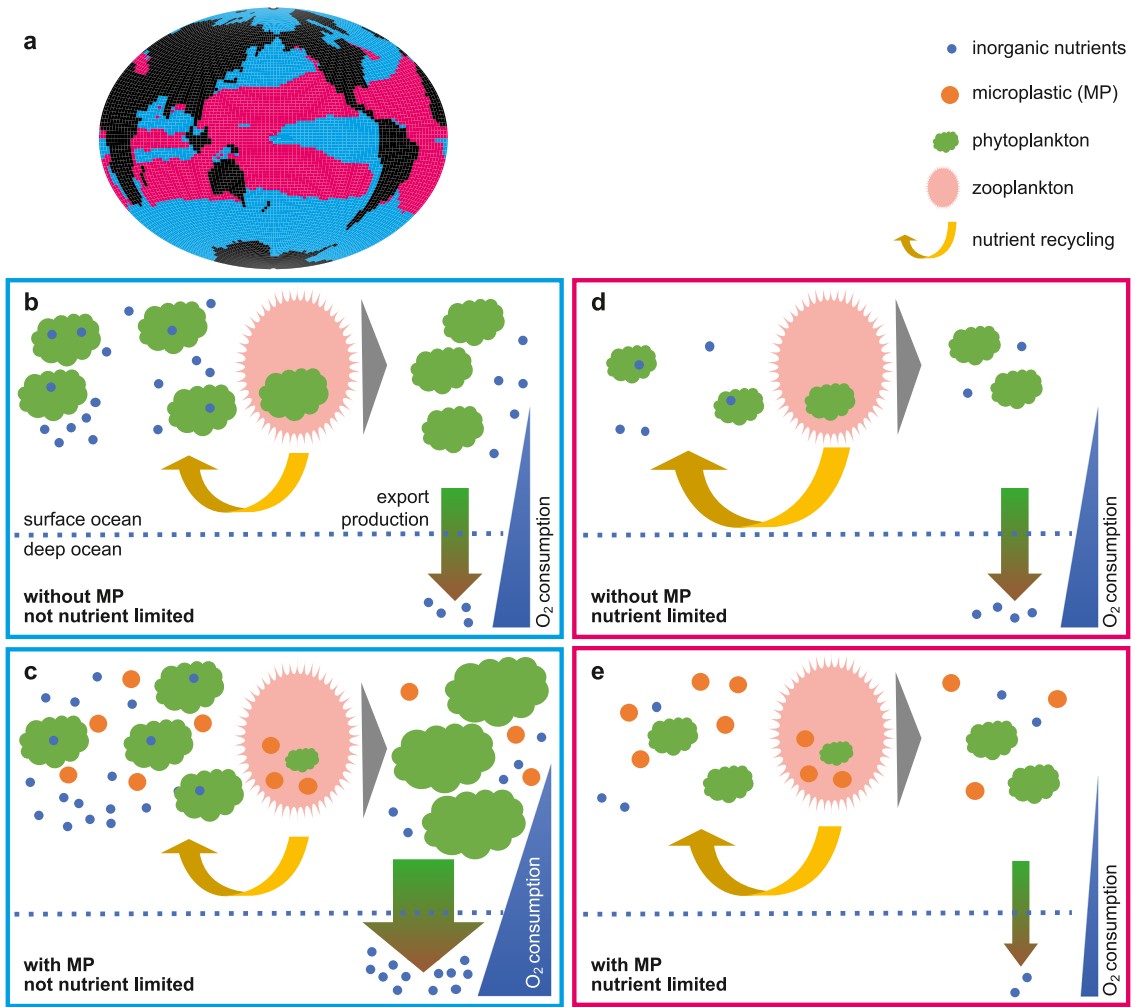

**Fig. 3 Schematic of impact of zooplankton ingestion of microplastic on water column dissolved oxygen. a** Map view of macronutrient limited (pink) and nutrient-replete regions (blue). **b** In macronutrient-replete regions, e.g. the Southern Ocean, grazing pressure from zooplankton is a significant control on primary and export production. **c** Consumption of microplastic by zooplankton in macronutrient-replete environments reduces the grazing pressure on primary producers enhancing export production, that upon remineralization at depth consumes oxygen. **d** In macronutrient-limited environments primary producers rely on recycled nutrients supplied via the microbial loop and zooplankton excretion. **e** In the presence of zooplankton ingestion of microplastic a greater proportion of nutrients cycles through the temperature-sensitive microbial loop, leading to a decrease in export production which in turn drives a reduction in oxygen consumption at depth for remineralization.

replete (Fig. 3). The net effect on biogeochemistry is a mix of macronutrient-limited and macronutrient-unlimited responses: enhanced surface nitrate depletion relative to Redfield in the Low Concentration and Moderate Concentration simulations, but not in the High Concentration simulation (Fig. 6), which suggests the possibility of zooplankton ingestion of microplastic increasing export production in the northern Indian Ocean on a seasonal scale (despite annually averaged decreased export production and increased water column oxygen).

**Southern Ocean response and global trend**. We find a systematic biogeochemical impact across the Southern Ocean, despite our models simulating low microplastic concentrations there. This impact occurs because the region is not macronutrient-limited (Fig. 3, panels b and c), therefore even slight alleviation of grazing pressure increases export production. The Southern Ocean responds similarly to the North Pacific, with an increase in export production due to the alleviation of grazing pressure that reduces water column oxygen inventory by as much as 15 mol $O_2$ m$^{-2}$ by 2020 across a wide geographical area

(Fig. 5), despite low microplastic surface concentrations. The significant losses of oxygen in the North Pacific and Southern Ocean dominate the enhancement of the global oxygen trend (Fig. 8, which presents the solution space of a model ensemble of 14 simulations that met the criteria described in the Methods section). By the year 2020, the additional global oxygen loss is between 0.2% and 0.5% relative to year 1960 values, which is significant considering the model simulates a climate-induced loss of oxygen of around 1% by this time due to a combination of solubility, circulation, and respiration effects[2]. Differences between simulations including microplastic grazing and the No Bio simulation grow with time, reflecting both climate change enhancement of the export trends as well as increased rates of microplastic pollution. By 2100, zooplankton grazing of microplastic can account for an additional 0.2-0.7% loss of global oxygen.

**Discussion**
The results presented here demonstrate the potential for zooplankton grazing on microplastic to have significant regional as

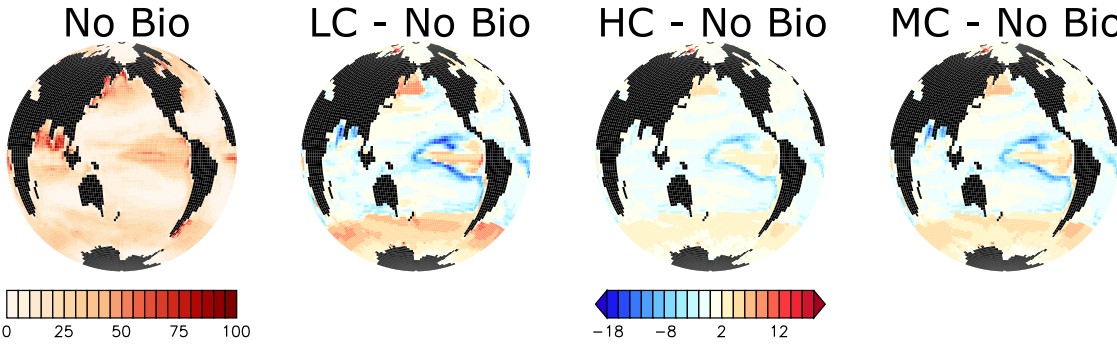

**Fig. 4 Simulated export production (g C m$^{-2}$ y$^{-1}$) at 130 meters depth at year 2020.** Differences between each of the three simulations that include biological uptake of microplastic and the No Bio simulation are shown together with absolute values for No Bio. Abbreviations are LC (Low Concentration), HC (High Concentration), and MC (Moderate Concentration).

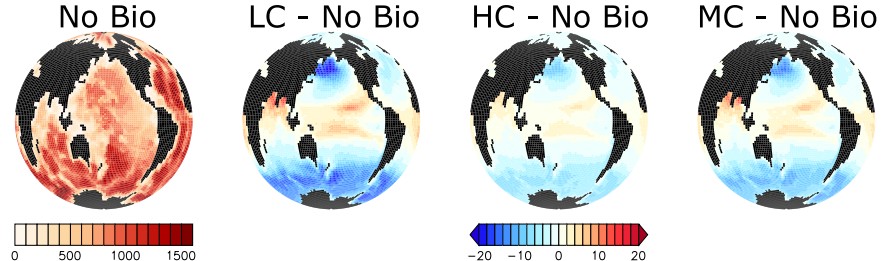

**Fig. 5 Simulated water column oxygen inventory (mol O$_2$ m$^{-2}$) at year 2020.** Differences between each of the three simulations that include biological uptake of microplastic and the No Bio simulation are shown together with absolute values for No Bio. Abbreviations are LC (Low Concentration), HC (High Concentration), and MC (Moderate Concentration).

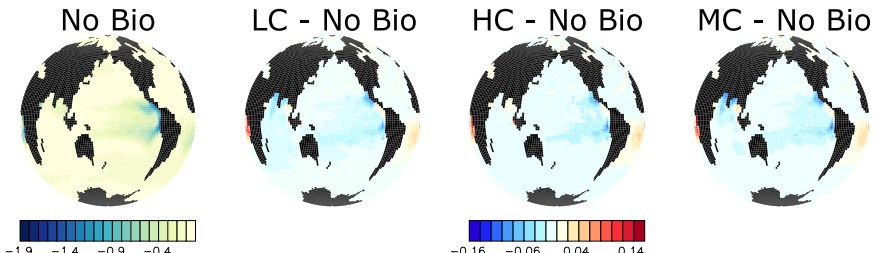

**Fig. 6 Simulated surface nitrate depletion relative to Redfield (1N:16P in mmol P m$^{-3}$) at year 2020.** Differences between each of the three simulations that include biological uptake of microplastic and the No Bio simulation are shown together with absolute values for No Bio. Abbreviations are LC (Low Concentration), HC (High Concentration), and MC (Moderate Concentration).

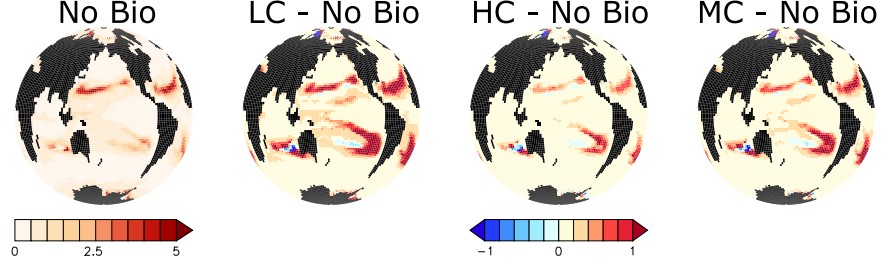

**Fig. 7 Simulated ratio of microbial loop recycling to zooplankton excretion, integrated over the top 130 meters at year 2020.** Differences between each of the three simulations that include biological uptake of microplastic and the No Bio simulation are shown together with absolute values for No Bio. Abbreviations are LC (Low Concentration), HC (High Concentration), and MC (Moderate Concentration).

well as downstream consequences on biogeochemical rates relevant for water column dissolved oxygen. These consequences can grow with time as microplastic accumulates in the ocean and exacerbate trends due to ocean warming and stratification, and are potentially significant enough to already be influencing the global deoxygenation trend.

We obtain these results using an intermediate-complexity earth system model that prescribes seasonally-cyclic iron limitation[15] for phytoplankton growth. However, in our model iron limitation is nowhere the primary limiting growth factor on annal average. Stronger simulated iron limitation might reduce the magnitude of ecosystem response in macronutrient-limited regions (e.g., blue

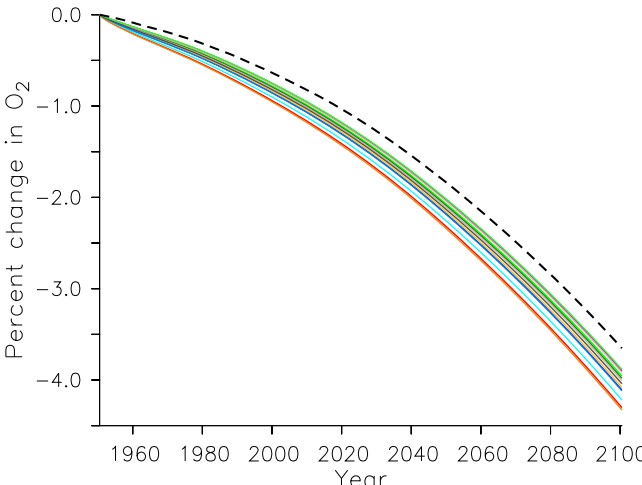

**Fig. 8 Timeseries of change in global oxygen inventory normalized to value at 1960.** The No Bio simulation is represented as a dashed black line, and the rest are the suite of 14 simulations that include biological interaction with microplastic, selected from a 300 member perturbed-parameter ensemble[7] that searched the parameter space of the model to find the most plausible microplastic uptake parameter combinations.

regions in Fig. 3 panel a). However, grazing pressure appears to be a key influence on particle export even in the iron-limited regions of high (macro)nutrients and low chlorophyll (HNLC regions, such as the northern latitudes of the Southern Ocean), as well as across the macronutrient-replete Southern Ocean[16]. A dynamic representation of iron in a more recent version of the model than used here indicates increasing surface iron concentrations with climate warming[17], which broadly reduces iron limitation over the next centuries. Nevertheless, a more careful examination of iron limitation's affect on the processes described in Fig. 3 is warranted.

Furthermore, the demonstrated sensitivity of the biogeochemical response to zooplankton relative grazing selectivity for the simulations presented here exceeds the sensitivity of the response to microplastic concentrations. This suggests both a potential for significant biogeochemical consequences already without significant surface microplastic accumulation, and an urgent need to constrain this very uncertain[5] parameter. Relative grazing selectivity for any given combination of microplastic, size or type and zooplankton group can be expected to vary, and values in our model represent a community response. They are thus difficult to constrain by observations and the model ensemble therefore encompasses a range of values that range from weaker to stronger selection relative to other food sources. Notably, even for simulations that use a relative grazing selectivity that favours phytoplankton over microplastic, accelerated global deoxygenation still occurs.

A lack of basic knowledge of plastic/biological interaction and microplastic distributions limits our conclusions to the hypothesis that even at the low concentrations simulated in our model, microplastic might affect biogeochemistry at a global scale (and might already be doing so). Whether the application of microplastic contamination of the food web to earth system model simulations of historical deoxygenation can improve the models' typically poor performance against observations[2] (and thereby offer compelling evidence of a missing mechanism in simulated earth system biogeochemistry) remains to be determined. Our study neglects other potential biogeochemical feedbacks that could be still more powerful, such as modified particle sinking rates[18] or zooplankton life cycle effects[5]. Nor does it explicitly consider multiple factors determining microplastic bioavailability

to zooplankton, e.g. multiple species, diel vertical migration and life stages, or the physical characteristics of the microplastic[5,19]. Which of these many factors that are known to be important for plastic/biological interaction as well as organic carbon export production at an individual level, may be relevant to pollution effects on biogeochemistry at a global scale is unknown. Given the large associated uncertainties, our results must be considered qualitative.

It has been hypothesised previously in the literature that microplastic contamination of the base of the marine food web may be significant enough to disrupt the biological pump[20–24] (albeit with mechanisms different from what we demonstrate here). Because of the numerous counteracting and non-linear effects, increasing export production in our microplastic model does not result in greater atmospheric $CO_2$ drawdown (anthropogenic carbon emissions diagnosed in our simulations are effectively equal). A more careful analysis of this question is left to the future, along with a host of other intriguing research questions, such as how the legacy of plastics in the environment might influence the centennial-scale adjustment of the biological pump to climate warming, or the quantification of biological pump or biogeochemical perturbation 'commitment' due to already released plastics. That plastics pollution might potentially alter carbon flux rates regionally produces still more unanswered questions regarding the effects on marine biomass-based climate mitigation measures and national and international carbon markets. We hope to develop answers to these questions in the future.

## Methods

**Experimental setup**. For this study we use model data generated in the writing of a companion manuscript published with Scientific Reports[7]. That article, and its accompanying Supplementary Information, provides full details of this Eulerian model and model forcings, as well as the complete model solution space exploration and an extensive analysis of idealised microplastic transport via the base of the ocean food web. The four simulations we describe here are the same as used in the main text of the companion article[7]. They were selected to represent the solution space of the 14 individuals that met a set of criteria, extracted from a 300 member Latin Hypercube[8] ensemble that explored the microplastic parameters' influence on biologically mediated transport. The Latin Hypercube ensemble explored the role of individual model parameters in biologically-mediated microplastic transport by repeating (with the model and model forcings described below) 300 variations of parameter values over a prescribed parameter range. In this approach, the model is not 'tuned' to match observations (which are very sparse for microplastic), but we attempt to identify regions of the parameter space with high/good skill at reproducing available observational estimates; in this case, global ocean microplastic inventory[9], using a plausible global ocean microplastic pollution rate[11], and marine snow aggregation fraction[12]. Fourteen individuals in this 300 member test simulated total ocean microplastic inventories that roughly agree with the independently calculated inventory at year 2010[9]. Four of them produced power-law shaped 'free' (unattached) microplastic concentration profiles consistent with recent observations from the North Pacific Gyre[25]. Local surface or sub-surface particle minima in the unattached partition are also present in 12 of the simulations, which broadly agrees with a 'missing' microplastic fraction near the surface[9], as well as observed intermediate-depth particle maxima for microplastic profiles found off California[10] and across the Atlantic Ocean[14]. In all 14 simulations, these characteristics were produced using plastic release[11] and marine snow aggregation rates[12] close to independent estimates. Parameter settings for the simulations selected for this article are given in the companion article[7], as well as in Table 1.

**Model description**. Our study uses the University of Victoria Earth System Climate Model (UVic ESCM) version 2.9[15,26–28]. The UVic ESCM is an intermediate-complexity earth system climate model with a resolution of 1.8° latitude by 3.6° longitude and 19 vertical depth levels in the ocean component. The surface ocean level is 50 m deep. The atmosphere component is a simple two dimensional energy-moisture balance model. Winds are prescribed from monthly NCAR/NCEP reanalysis data and are geostrophically adjusted to surface pressure changes based on temperature anomalies[26]. Terrestrial carbon, ocean circulation, sea ice, and ocean sediments are represented dynamically. Ocean biogeochemistry is represented with a relatively complex nutrients-phytoplankton-zooplankton-detritus model that includes three phytoplankton functional types (diazotrophs, calcifiers, and general phytoplankton) and one zooplankton type[28]. Biogeochemical model pre-industrial

steady-state has been previously described[28], as has model response to climate forcing[29]. Model biogeochemistry has been additionally modified to separate faecal pellets from general detritus, but both are prescribed equal particle sinking rates (unchanged from previously published model versions) so that this modification does not influence biogeochemistry[7]. Marine snow production is calculated as a fraction of general detritus production. Marine snow aggregates are diagnosed for the purposes of applying the microplastics model, but are not explicitly traced[7].

A microplastics component has been implemented in the ocean module[7]. Three microplastics tracers are introduced. These tracers do not resolve different polymer types, nor do they resolve particle size or abiotic degradation. Thus, our 'microplastic' compartments must be considered generic, biologically-interactive particles that affect the base of the marine food web. The base unit of all microplastic tracers is particles per cubic meter. 'Free' microplastic is unattached to biological particles. It is introduced to the ocean from coastlines and shipping lanes and mixes passively via advection and diffusion. A fraction of the concentration in each grid cell is assigned a very fast rise rate to represent some particles having positive buoyancy. This fraction is subjected to sensitivity testing in our companion article[7]. There is no direct sink of free microplastic and no abiotic degradation of the particles, so free particles released to the ocean will remain in the ocean unless removed by biological aggregation and sinking. Microplastic bound in marine snow is the second microplastic tracer. Free microplastic aggregates with a fixed fraction of the general detritus produced to produce marine snow/microplastic aggregates. The associated parameters are subjected to sensitivity testing[7]. Biofouling is implicitly represented by the microbial loop in our model, which breaks down organic material (including microplastic-laden marine snow aggregates) and releases the microplastic back into the 'free' partition. Microplastic bound in marine snow sinks at the rate of detritus. A fraction of the marine snow-bound microplastic particles that sink to the seafloor are considered to be lost from the ocean via sedimentary burial, whereas the remainder is returned to the 'free' partition in the bottom ocean grid cell. This fraction is subjected to sensitivity testing[7]. Zooplankton graze 'free' microplastic particles- for simplicity, they do not consume microplastic held in marine snow aggregates. Unique grazing selectivity for each food type (free microplastic, detritus, phytoplankton types, zooplankton) that add to one are prescribed by expanding the Holling II grazing formulation to include microplastic grazing. Changing the Holling II formulation would require an entire model assessment and parameter sensitivity from scratch, so would not be an option for this study. A range of microplastic grazing selectivity values, from relative aversion (i.e. a smaller grazing selectivity value relative to other food sources) to a high relative selectivity, are tested[7]. Adjustment of the relative grazing selectivity implicitly accounts for effects such as particle size, occurrence of accidental ingestion, or biofouling affecting prey (microplastic) rejection[30]. The formulation of microplastic grazing rate ($G_{MP}$; in units of microplastic particles per cubic metre per second) is:

$$G_{MP} = \mu_Z^{max} Z \frac{\psi_{MP} MP\ R_{M:P}\ R_{F:MP}\ R_{N:F}}{\psi_{CO} CO + \psi_{PH} PH + \psi_{DZ} DZ + \psi_{Detr_{tot}} Detr_{tot} + \psi_Z Z + \psi_{MP} MP\ R_{M:P}\ R_{F:MP}\ R_{N:F} + k_Z}$$

(1)

The maximum potential grazing rate ($\mu_Z^{max}$) is scaled by zooplankton population (Z) and microplastic concentration (MP), and weighted by the relative food selectivity ($\psi_{MP}$), the availability of alternative prey (CO, PH, DZ, Detr$_{tot}$), and Z representing organic food sources[29] and a half saturation constant for zooplankton ingestion ($k_Z$). Relative grazing selectivities must always sum to 1. Therefore, a lower relative grazing selectivity for e.g. microplastic, requires an increase in the relative grazing selectivity of other food sources. We set $\psi_{DZ}$ to 0.1 for all simulations (on the basis that diazotrophs are a poor food source, and to minimize disruption to the nitrogen cycle). Microplastic is for ingestion considered as food source in analogy to alternative sources (e.g. CO), but with no nutritive benefit to the zooplankton. To calculate biological uptake, microplastic particles are converted to grams using the microplastic particle-to-mass conversion of 236E3 tonnes MP = 51.2E12 particles MP ($R_{M:P}$)[31]. We assume that 1 g of microplastic will roughly replace 1 g of food (at Redfield ratios; $R_{N:F}$ is the conversion from mol Food to mol N) in the zooplankton's diet, and microplastic is thus converted to mmol N for the grazing calculation. This ratio ($R_{F:MP}$) is subjected to sensitivity testing[7].

For simplicity, we assume 100% plastic particle egestion efficiency (niether microplastic remains in the zooplankton gut, nor is it metabolised). Microplastic particles bound in faecal pellets sink at the rate of detritus, and are released back to the 'free' partition by bacterial remineralisation of the organic matrix. Like marine snow-bound particles, a prescribed fraction that reaches the seafloor is considered to be lost from the sea and the remainder is released back to the 'free' partition in the bottom grid cell. The loss of biologically-transported microplastic particles to the seafloor can significantly reduce total ocean particle inventory[7]. Likewise, marine snow uptake of microplastic strongly controls particle distributions in the surface[7], and both of these factors can help to determine the bioavailability of microplastic for the zooplankton (hence, marine snow aggregation is important to resolve in our simulations of zooplankton grazing of microplastic). Microplastic particle size is not simulated in our study, which might bias our results to the lower end of the defined microplastic size range. However, model parameters do implicitly consider particle size effects, e.g. the relative grazing selectivity and the food:microplastic substitution ratio.

**Model forcing**. The model was integrated at year 1765 boundary conditions (including agricultural greenhouse forcing and land ice) for >10,000 years until equilibration was achieved. From year 1765 to 1950, historical $CO_2$ forcing, and geostrophically adjusted wind anomalies are applied. From 1950 to 2100 the model is forced with a combination of historical $CO_2$ forcing (to 2000) and a business-as-usual high atmospheric $CO_2$ concentration projection (RCP8.5)[32,33]. Model physical response is identical across model configurations, therefore biogeochemical differences arise purely due to differences in the application of zooplankton consumption of microplastic. Microplastic emissions start from 2 million metric tonnes in year 1950 (an annual total plastic production estimate[34]), increasing at a rate of 8.4% per year. Microplastic emissions are weighted spatially by coastal $CO_2$ emissions (stationary as well as emissions along major shipping lanes) calculated for the 1990s[7]. It has been estimated that 4% of total plastic waste generated enters the ocean[11]. We apply a range of input fractions (see Table 1 and[7]), after applying a mass conversion from tonnes to number of microplastic particles[31].

## Data availability
Model output used for writing this manuscript has been assigned the digital handle 20.500.12085/a3c47dd2-ee93-475f-bca8-e7f58d3c2fec. It is available at: https://data. geomar.de/downloads/20.500.12085/a3c47dd2-ee93-475f-bca8-e7f58d3c2fec

## Code availability
Model code is available using the same digital handle as the model output and is provided without restriction at: https://data.geomar.de/downloads/20.500.12085/a3c47dd2-ee93-475f-bca8-e7f58d3c2fec.

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

## Acknowledgements
The authors wish to acknowledge computing resources made available by GEOMAR Helmholtz Centre for Ocean Research, Kiel, and Kiel University. K.K. would also like to acknowledge topical discussion with S. Mittenthal and W. Yao. Figures made in the paper were produced using the Ferret plotting program and by Rita Erven (Fig. 3).

## Author contributions
K.K. wrote the model and conducted the simulations in consultation with A.E.F.P. and A. O. C.-T.C and A.L. contributed the Latin Hypercube framework. K.K. wrote the paper. All authors contributed to the model analyses and editing of the paper.

## Funding

## Competing interests
The authors declare no competing interests.
