## [Peer Review File · Nature Communications]

REVIEWER COMMENTS

Reviewer #1 (Remarks to the Author):

This manuscript describes the possible effect of microplastics on the export of organic carbon from the surface ocean and the consequent decrease in oxygen concentration in the deep ocean. The conclusions result from a model of the effects of microplastics on the upper ocean carbon cycles. This is definitely a novel model result, and most likely not something that most oceanographers have considered. For that reason it is an interesting study and an important new direction.

A deep understanding of the result, however, is going to be difficult because all details of how the model works are described in another paper that is presently in review in Scientific Reports. Because of the very short nature of this communication one will know almost nothing about the model details without reading the longer paper when and if it is published.

The shorter paper focuses on one aspect of the plastics problem which is the effect on the biological pump and oxygen concentration. It reads as if the authors took a few figures from the longer paper and wrote a short communication around them. The oxygen concentration in the model decreases by about 1% from 1960 to 2020 without plastics and another 0.2 – 0.5 % with the effect of plastics. So, grazing of microplastics makes a significant difference on the rate of deoxygenation, and that is the point of the paper. The reason for the model decrease in oxygen concentration even without microplastics is not described and neither is how well this result compares with data.

The clear weakness of paper conclusions is that the oceanographic community knows so little about how microplastics affect grazers that it is impossible to know how serious to take the model results. The authors pretty much acknowledge this in the last paragraph before the Methods. This is my main criticism of the present paper, but there are a few smaller issues that are enumerated below.

1. The very beginning of the manuscript is a disputable point. The first sentence says, "Global warming has driven a loss of dissolved oxygen in the ocean over the past decades." Measurements have shown that the oxygen has decreased, but we do not know that it is entirely a global warming effect. There could be long-term cycles we do not understand!
2. What the heck is a "Latin Hypercube simulation"?
3. I find the results in Figure 1 very difficult to follow because differences are due to both the uptake by marine snow and variable plastics/plankton grazing pressure. I think it is nearly impossible for the reader to know what are the most important influences when more than one aspect of the model is changed.
4. Are the 20 maps in Fig 2 all necessary, or could the main point be made with just a few?
5. The different lines in Fig. 4 are not labeled.
6. In the very last sentence it says, "40% of the total plastic production enters the ocean.." I think it is more like 4%. This must be a typo??

Reviewer #2 (Remarks to the Author):

In this paper, authors analyse the mechanisms of microplastic spread in the ocean by using Lagrangian particle models. They focus on the plastic accumulation in the ocean and the novelty of their research is that the authors take into account the biological transportation of MP particles, as it can change their distribution in the ocean. In the paper, they created some interesting simulations using historical data of CO₂ and making predictions until 2100.

The research questions are of interest and the methodology is novel, but the data presented are poorly presented and they hide potentially interesting findings relating the MP accumulation in the ocean. The interpretation and related discussion do not always support the results. The supplementary Information give better understanding of the interpretation of the data, more than the main text. I believe that there is need for further research on the hypotheses for microplastic interaction with ocean biology. The points below highlight the keys areas of concern and/or where further information

and analysis would be necessary to support authors' statement;

Comments:

Abstract

The abstract is not currently clear. I could not relate the abstract with the rest of the text.

"about four percent of the plastic waste generated worldwide ends up in the ocean" -- Based on Greenpeace report, of the 260 million tons of plastic the world produces each year, about 10 percent ends up in the Ocean (Plastic Debris in the World's Oceans, 2006) and the seventy percent of the mass eventually sinks, damaging life on the seabed.

"buoyant larger plastics" -- Vague. What size?

"the smallest size fractions (the microplastics)" -- Vague. Very ambiguous and not at all clear.

Introduction

Line 12-13; "Large, floating plastic detritus has historically captured the largest share of attention by the scientific community, the media, and the public." -- Vague. References should be added. What is the size of a "large, floating plastic detritus"?

Line 14; "a substantial "missing" fraction of plastic" -- What do you mean "missing"? It is not clear the term.

Line 20; Tiny plastic particles missing -- What do you mean "tiny"? What size is tiny? Ambiguous.

Table 1; Probably should be moved in Method section. It is not clear how it is related to the introduction section.

Line 67-70? ; "Explicit modelling of biological interactions with MP ... into the future" -- Could you give more details on this advantage?

Results

Line 75; "greatly" -- This is a subjective term.

Biology offers microplastic a path to the deep sea

Line 105, 234, 276, Figure 7, Table S1; "total annual waste generation" -- What is the total annual waste generation? Globally or locally?

Line 106; "the smallest fraction" -- What is the smallest fraction? Vague.

Line 107; "lower overall particle inventory and smallest surface concentrations" -- What number is the lower overall particle and the smallest surface concentrations? Confused.

Line 107 & 110; "the other two models." -- What are other two models. Which ones?

Discussion

Line 203-204; "modification of particle sinking rates, higher trophic levels, plastic polymer type and plastic particle size." -- Why these were excluded?

Line 204-205; "The particle size is relevant to the formation of marine aggregates, which mostly incorporate plastics smaller than 1 mm, and to size-selection by zooplankton." -- What is the size-selection?

Line 206; "polymer type" -- What polymer type? How do we know that?

Line 214; "energy moisture-balance model" -- Typo: Energy-moisture balance model

Model description

Line 229; "Plastic particles have been observed to both increase and decrease the sinking rates of marine snow and decrease the sinking rates of faecal pellets" -- What are the sinking rates? Authors seem to have this information, but it is not given to the readers.

Line 230-231; "All MP are considered to represent particles within a biologically active size range" -- The sentence is unclear.

Page 8; "but this size (and the particles' composition) is never made explicit." -- Ambiguous and no clear sentence.

Line 251-252; "What fraction of MP particles reaching the seafloor via aggregate and faecal pellet ballasting are returned to the water column (FB) is tested." -- How was it tested?

Line 254; "The calculation of MP particle ingestion rate (Pupt 254) is the same as for other food sources." -- What is value for the ingestion rate?

Line 278; "the microplastic mass" -- How much mass?

Line 280-281; "after applying a mass conversion from tons to number of MP particles. Using a considerable over-estimation of MP pollution rate also implicitly accounts for abiotic degradation of larger plastics." -- Ambiguous. Why did you apply mass conversion? And for what reasons did you decide to use over-estimation of MP pollution rate? What is that rate?

MP with no biological uptake stays in the upper ocean

Line 474; "total emitted MP inventory" -- How much is the total MP inventory?

Line 472; "total ocean MP inventory (second row, Fig. S1) and maximum surface concentration in gyres" -- How much is the maximum surface concentration in gyres?

Line 480; "total passive inventory" -- How much is total passive inventory?

Latitudinal MP distributions are sensitive to the fraction assigned a rise rate

Line 487; "applied buoyancy assumptions" -- How then is it possible to have accurate results if they are assumptions? What are those assumptions?

thredds.geomar.de/thredds/catalog/open_access/kvale_et_al_2020_npg/catalog.html -- Appears that message "HTTP Status 404 - Not Found; THREDDS Data Server Version 4.6 -- Documentation"

References

8. Day, R. H. & Shaw, D. G. Patterns in the abundance of pelagic plastic and tar in the north pacific ocean, 1976–1985. *Mar.Pollut. Bull.* 18, 311 – 316, DOI: [https://doi.org/10.1016/S0025-326X\(87\)80017-6](https://doi.org/10.1016/S0025-326X(87)80017-6) (1987). -- The DOI does not work.

Lindeque, P. K. et al. Are we underestimating microplastic abundance in the marine environment? a comparison of 355 microplastic capture with nets of different mesh-size. *Environ. Pollut.* 265, 114721, DOI: <https://doi.org/10.1016/j.envpol.2020.114721> (2020). -- The DOI does not work.

Wichmann, D., Delandmeter, 359 P. & van Sebille, E. Influence of near-surface currents on the global dispersal of marine 360 microplastic. *J. Geophys. Res. Ocean.* 124, 6086–6096, DOI: 10.1029/2019JC015328 (2019). <https://agupubs.onlinelibrary.wiley.com/doi/pdf/10.1029/2019JC015328>. -- The link does not work. When you click it, it does not take the whole link as one line.

Zhu, L., Zhao, S., Bittar, T. B., Stubbins, A. & Li, D. Photochemical dissolution of buoyant microplastics to dissolved 391 organic carbon: Rates and microbial impacts. *J. Hazard. Mater.* 383, 121065, DOI: <https://doi.org/10.1016/j.jhazmat.2019.392> 121065 (2020). -- The DOI does not work.

Page 14 -- The references have a larger gap among them.

Points to be considered;

There is an excessive use of characterisations of “large”, “small”, “tiny”. You clarified the size of MP in your introduction (MP; 0.1 to 5 mm in diameter). These adjectives make the reader to think that the particles are larger or smaller the referred size of MP. If this is the case, there is the need to clarify the size.

Did you take into consideration the disadvantages of using UVic ESCM?

an absolute surface temperature that’s a bit too low

projections of ocean heat uptake that are a bit too high

In what level these disadvantages can influence the interpretation of the data?

Figures, the numbers are small and blurry. The figures should be in bigger size in order to be easily readable.

Finally, I would like to recommend to the authors the article below:

“Bioavailability and effects of microplastics on marine zooplankton”, Zara L.R. Botterell, Nicola Beaumont, Tarquin Dorrington, Michael Steinke, Richard C. Thompson, Penelope K. Lindeque, <https://www.sciencedirect.com/science/article/pii/S0269749118333190>

Reviewer #3 (Remarks to the Author):

Review of Kvale et al. "Zooplankton grazing of microplastic can accelerate global loss of ocean oxygen"

This study describes an interesting new potential pathway of how oxygen concentration in the ocean could be affected by microplastics, a hitherto unknown process. I like the idea – provocative and important. Who knows whether it is correct, but it is logical and in my view deserves more attention in the future. However, there are a number of comments that the authors should address, mainly around biological realism and better articulating the model limitation and explaining things, which will make the study stronger:

1. Note that as you define microplastics (down to 0.1 mm), zooplankton eat very few particles in this size range. Most of the phytoplankton zooplankton eat are much smaller than this. Further, not many NPZD models or ESMs model phytoplankton >100 um in size (virtually none?). Maybe you need to be talking about mini-microplastics down to 5 um in size (Brandon et al. 2019)?
2. You say in the Abstract "Although significant uncertainty accompanies these estimates,". Fig. 4 suggest that you have very little uncertainty in your model. I suspect there is no structural uncertainty in your ESM wrt phytoplankton or zooplankton functional groups. It is not clear even how many functional groups of zooplankton are in the model – is there one or two? I'd like to see some discussion of the caveats of the limited structural uncertainty in the model. I understand that ESMs have very basic zooplankton (and phytoplankton), but your results could be quite different depending on the zooplankton group (e.g. copepods, larvaceans, salps, krill or jellyfish – herbivores or carnivores) that were present. This should all be in a caveats section
3. On the theme of what should be mentioned in a caveats section, I'd like to see more discussion of this. You are focusing on export production and oxygen concentration – what about diel vertical migration, different sinking rates of different zooplankton functional groups, marine snow formation by larvaceans? All of these can affect export production and oxygen concentration in deeper layers, but I suspect none are in the model.
4. I'd like to see some justification for why you have included marine snow? Zooplankton eat microplastics if they are not part of marine snow. And only some zooplankton eat marine snow, whilst others eat phytoplankton. More justification is needed about the model structure
5. The main reference you use for convincing the reader that zooplankton ingest microplastics is Cole et al. (2013). Can you cite some more recent work as well that confirms that zooplankton eat microplastics?
6. Little evidence is provided about the selectivity of microplastics by zooplankton. In the study you run a range of simulations from low to high ingestion, but I'd like to see some references showing that this is what happens in zooplankton. For example, is there evidence of zooplankton preferring microplastics to phytoplankton/marine snow (i.e. your TestLow scenario)
7. One piece of evidence you might like to include explaining why zooplankton ingest microplastics is that a biological layer colonises the particle, so that zooplankton think it is biological in nature (e.g. Dudek et al. 2020 *Limnology and Oceanography Letters*)
8. I think your message would be more effective if you were not talking in model speak with some of your terms. It is not reader-friendly using terms such as TestLo, TestHi, TestMed, NoGraz. This is throughout the figures too. Please make it easy on the reader and use more descriptive terms. You can make it longer to make it easier
9. You should be consistent in describing the zooplankton feeding preference. Sometimes you say "zooplankton feeding preference for marine snow" and other times you say "zooplankton feeding preference for microplastics". This is likely to confuse readers. Please be consistent
10. Fig. 3. I like the idea but think it can be executed better. Have a key explaining the symbols. The typesetting for A. and D. are poor and it seems like the export is going into them rather than the deep ocean?
11. A logical step in your explanation is missed in the text and in Fig. 3. With zooplankton ingesting microplastics in nutrient replete system, reduces grazing pressure on phytoplankton and then leads to more export production, misses the step of phytoplankton increase in between. More explanation is needed to help the reader. This is especially true because the phytoplankton don't really increase in nutrient-deficient areas because they rely on recycled nutrients. It will help the reader understand if

you provide more description of the process.

12. I like the idea of Fig. 3, but feel it could be improved and made clearer. For example, I don't find the reduction in the recycling by zooplankton is included in the Not nutrient limited systems. Would a sequence of panels for each of the Nutrient limited and Not nutrient limited systems better describe the changes in the system that happen?

13. You say "At a global scale, the additional loss of oxygen in the regions not limited by nutrients exceeds the gain of oxygen in regions limited by nutrients." It is very difficult to see the relative size of different areas in your plots because it appears you have not used a projection. Please use a projection in your figures that better represents the area of each ocean

14. In the regions that are nutrient replete and show an increase in primary production, will there be an increase in carbon dioxide drawdown from the ocean to the atmosphere? i.e. Does your work suggest the biological pump be stronger? Please comment

15. Figure 4 needs axis labels

16. eds to be more description of the model – how many phytoplankton groups, how many zooplankton groups. There is no reason why at least the very basics of the biogeochemical part of the ESM could not be explained.

17. I'd like to see some suggestions of where to go next with this work? What are the next steps?

18. Figure 4 needs axis labels

19. Fig. 2. I don't understand. What does No Graz mean everywhere? Please explain

20. There are relatively few references in this study, but that might be a limitation of the journal. I found quite a few statements needed support by reference and didn't have it.

Anthony J. Richardson

Reviewer #4 (Remarks to the Author):

Kvale et al present an ambitious modelling-centric paper that looks to link laboratory derived data on zooplankton to global oxygen loss in response to microplastics. Unfortunately, the paper has made too many leaps, statements are vague, many steps have incorrectly selected/understood data from other sources, and lacks transparent methodology. I cannot recommend this paper for publication. Additional comments below:

Title - misleading, as in Fig 2E the authors show oxygen loss and decline depending on latitude
L15 - a lower bound of 0.1 mm = 100 micron which is above size range consumed by many zooplankton; lower range would be 0.001 mm.

L19 and elsewhere - Throughout the paper there is heavy reference to reference 6, coined as a companion MS. This has not been peer-reviewed as yet, and this reviewer has no way to know whether the methodology used is appropriate.

"significant influence of marine snow and zooplankton fecal pellets" -- this is a big statement that cannot be checked owing to reference 6 not being available; but there are no caveats given either, is this based on experimental data only? such experiments often have caveats of very high MP numbers, single type of MP etc. these caveats and errors are not carried forward sufficiently.

Test Lo/Med/Hi are referred to in text, but these are arbitrary. There is no data or indication of the extent of the low/high settings and whether they're in anyway realistic. Fig 1 legend doesn't state what these codes properly refer to (i.e. marine snow influence).

Fig 1 text in axis is too small to read.

L39 "Microplastic offers zooplankton an alternative food source" - no, from what I have read it is captured alongside their food.

L57 - "depending on model configuration" - be specific.

L62 - zooplankton grazing preference for MP? I have seen no evidence that they would selectively graze on MP over algae, quite the opposite.

Methods - massive lack of transparency here, with no detail given.

Better justification of microplastic loadings are required; why start at 1950 for "micro"plastic? How

were the microplastic levels in the model comparing with environmental levels actually observed. This is never stated in text and I don't feel the MP loads are elucidated in the figures. How much plastic is causing the observed shifts in the models?

L127 - "40% of total plastic production enters the oceans"... no, this is factually incorrect. The Jambeck paper consider a hypothetical extreme of 40% of mismanaged waste from coastal populations 50km from coasts might end up in ocean, but this is not the same as you state.

Reviewer #1 (Remarks to the Author):

This manuscript describes the possible effect of microplastics on the export of organic carbon from the surface ocean and the consequent decrease in oxygen concentration in the deep ocean. The conclusions result from a model of the effects of microplastics on the upper ocean carbon cycles. This is definitely a novel model result, and most likely not something that most oceanographers have considered. For that reason it is an interesting study and an important new direction.

A deep understanding of the result, however, is going to be difficult because all details of how the model works are described in another paper that is presently in review in Scientific Reports. Because of the very short nature of this communication one will know almost nothing about the model details without reading the longer paper when and if it is published.

The authors would like to thank the reviewer for their careful reading of our manuscript and thoughtful suggestions, which have helped us to improve its quality. The Scientific Reports manuscript has now been published (<https://www.nature.com/articles/s41598-020-72898-4>). We have also now expanded our Methods section to include more detail about the model and modelling assumptions. Please see the changes to the manuscript made in blue font, which address this and the other concerns listed.

The shorter paper focuses on one aspect of the plastics problem which is the effect on the biological pump and oxygen concentration. It reads as if the authors took a few figures from the longer paper and wrote a short communication around them. The oxygen concentration in the model decreases by about 1% from 1960 to 2020 without plastics and another 0.2 – 0.5 % with the effect of plastics. So, grazing of microplastics makes a significant difference on the rate of deoxygenation, and that is the point of the paper. The reason for the model decrease in oxygen concentration even without microplastics is not described and neither is how well this result compares with data.

Climate drivers of deoxygenation have been explored extensively in earth system models, including this one. We reference a synthesis in the Introduction (Oschlies et al., 2018). New text is added to the Results section “Southern Ocean response and global trend” to clarify the reasons for climate-induced deoxygenation (Line 121).

Large uncertainties in our modelling framework discourage direct comparison to timeseries of O₂ data. For example, our coastal plastic pollution input rates are probably very inaccurate with respect to their spatial weighting, and this might affect the deoxygenation pattern. Spatial patterns and magnitudes of observed deoxygenation are now well captured by Earth system models (including ours) (Oschlies et al., 2018). Whether the inclusion of microplastic contamination can *improve* the deoxygenation pattern is a question left for future study, once we can improve the constraints on our model. What we do demonstrate here, as a first step, is that it is *possible* for plastic to significantly alter biogeochemistry at a global scale. Discussion of this point is now expanded in the Discussion section (Line 139).

The clear weakness of paper conclusions is that the oceanographic community knows so little about how microplastics affect grazers that it is impossible to know how serious to take the model results. The authors pretty much acknowledge this in the last paragraph before the Methods. This is my main criticism of the present paper, but there are a few smaller issues that are enumerated below. We agree that large uncertainties remain. However, by demonstrating that microplastics may substantially affect biogeochemistry we are hoping to bring this topic onto the research agenda of a wider community to be able to reduce unknowns and better constrain our models and test the hypotheses put forward here with new observations in the future.

1. The very beginning of the manuscript is a disputable point. The first sentence says, “Global warming has driven a loss of dissolved oxygen in the ocean over the past decades.” Measurements have shown that the oxygen has decreased, but we do not know that it is entirely a global warming effect. There could be long-term cycles we do not understand!

Observations suggest that deoxygenation trends over the past decades are ubiquitous (Helm et al., 2011, Schmidtko et al., 2017), which indicates a common global driver (not a regional oscillation). Furthermore, global deoxygenation is a clear effect of climate warming and is consistently demonstrated in earth system models (Oschlies et al., 2018). We agree

that the observed oxygen changes are not always and everywhere entirely caused by global warming and may well be moderated by long-term fluctuations, but we do not think that it is disputable (or disputed) that global warming has driven a loss of oceanic oxygen, as indicated by our sentence.

2. What the heck is a “Latin Hypercube simulation”?

We apologize for the jargon. Latin Hypercube sampling is a method for generating semi-random samples of parameter values from multiple dimensions. The method is now better described, and a citation provided (Line 165).

3. I find the results in Figure 1 very difficult to follow because differences are due to both the uptake by marine snow and variable plastics/plankton grazing pressure. I think it is nearly impossible for the reader to know what are the most important influences when more than one aspect of the model is changed.

Microplastic transport is also controlled by marine snow uptake, but what our results show is that the loss of ocean oxygen depends less on how much plastic is available at the surface and more on how much zooplankton prefer to eat it. This is one of the major findings of our study and we have emphasised this more both where Figure 1 is discussed (Line 46), and by expanding the Discussion (Line 129). A detailed description of the individual model parameters and their influence on microplastic distributions is provided in the now-published Scientific Reports article as Supplemental Information (<https://www.nature.com/articles/s41598-020-72898-4>), and we highly recommend it to anyone curious about the transport effects of parameters.

4. Are the 20 maps in Fig 2 all necessary, or could the main point be made with just a few?

The figure has been split into multiple figures, but we do feel all are important to demonstrate the points made in the text.

5. The different lines in Fig. 4 are not labeled.

The point with this figure is to show the solution space of our multiple simulations each with different parameter combinations. Hence labelling of each simulation may be confusing and somewhat meaningless. This is now made more explicit in the text (Line 118).

6. In the very last sentence it says, “40% of the total plastic production enters the ocean..” I think it is more like 4%. This must be a typo??

We apologise for this error and have corrected the estimate.

Reviewer #2 (Remarks to the Author):

In this paper, authors analyse the mechanisms of microplastic spread in the ocean by using Lagrangian particle models. They focus on the plastic accumulation in the ocean and the novelty of their research is that the authors take into account the biological transportation of MP particles, as it can change their distribution in the ocean. In the paper, they created some interesting simulations using historical data of CO₂ and making predictions until 2100.

The research questions are of interest and the methodology is novel, but the data presented are poorly presented and they hide potentially interesting findings relating the MP accumulation in the ocean. The interpretation and related discussion do not always support the results. The supplementary Information give better understanding of the interpretation of the data, more than the main text. I believe that there is need for further research on the hypotheses for microplastic interaction with ocean biology. The points below highlight the keys areas of concern and/or where further information and analysis would be necessary to support authors' statement;

The authors would like to thank the reviewer for carefully considering our manuscript, and for offering constructive comments. The review seems to focus on the Scientific Reports manuscript, now published with the majority of their criticisms expressed here addressed in the published version of that manuscript. We have tried to accommodate their relevant criticisms (shown included in the orange font changes to the manuscript), and also included reference to their final suggested manuscript, as that one does apply to this study (Botterrell et al., 2019).

Comments:

Abstract

The abstract is not currently clear. I could not relate the abstract with the rest of the text.

“about four percent of the plastic waste generated worldwide ends up in the ocean” -- Based on Greenpeace report, of the 260 million tons of plastic the world produces each year, about 10 percent ends up in the Ocean (Plastic Debris in the World's Oceans, 2006) and the seventy percent of the mass eventually sinks, damaging life on the seabed.

“buoyant larger plastics” -- Vague. What size?

“the smallest size fractions (the microplastics)” -- Vague. Very ambiguous and not at all clear.

Introduction

Line 12-13; “Large, floating plastic detritus has historically captured the largest share of attention by the scientific community, the media, and the public.” -- Vague. References should be added. What is the size of a “large, floating plastic detritus”?

Line 14; “a substantial “missing” fraction of plastic” -- What do you mean “missing”? It is not clear the term.

Line 20; Tiny plastic particles missing -- What do you mean “tiny”? What size is tiny? Ambiguous.

Table 1; Probably should be moved in Method section. It is not clear how it is related to the introduction section.

Line 67-70? ; “Explicit modelling of biological interactions with MP ... into the future” -- Could you give more details on this advantage?

Results

Line 75; “greatly” -- This is a subjective term.

Biology offers microplastic a path to the deep sea

Line 105, 234, 276, Figure 7, Table S1; “total annual waste generation” -- What is the total annual waste generation? Globally or locally?

Line 106; “the smallest fraction” -- What is the smallest fraction? Vague.

Line 107; “lower overall particle inventory and smallest surface concentrations” -- What number is the lower overall particle and the smallest surface concentrations? Confused.

Line 107 & 110; “the other two models.” -- What are other two models. Which ones?

Discussion

Line 203-204; “modification of particle sinking rates, higher trophic levels, plastic polymer type and plastic particle size.” -- Why these were excluded?

Line 204-205; “The particle size is relevant to the formation of marine aggregates, which mostly incorporate plastics smaller than 1 mm, and to size-selection by zooplankton.” -- What is the size-selection?

Line 206; “polymer type” -- What polymer type? How do we know that?

Line 214; “energy moisture-balance model” -- Typo: Energy-moisture balance model

Model description

Line 229; “Plastic particles have been observed to both increase and decrease the sinking rates of marine snow and decrease the sinking rates of faecal pellets” -- What are the sinking rates? Authors seem to have this information, but it is not given to the readers.

Line 230-231; “All MP are considered to represent particles within a biologically active size range” -- The sentence is unclear.

Page 8; “but this size (and the particles’ composition) is never made explicit.” -- Ambiguous and no clear sentence.

Line 251-252; “What fraction of MP particles reaching the seafloor via aggregate and faecal pellet ballasting are returned to the water column (FB) is tested.” -- How was it tested?

Line 254; “The calculation of MP particle ingestion rate (Pupt 254) is the same as for other food sources.” -- What is value for the ingestion rate?

Line 278; “the microplastic mass” -- How much mass?

Line 280-281; “after applying a mass conversion from tons to number of MP particles. Using a considerable over-estimation of MP pollution rate also implicitly accounts for abiotic degradation of larger plastics.” -- Ambiguous. Why did you apply mass conversion? And for what reasons did you decide to use over-estimation of MP pollution rate? What is that rate?

MP with no biological uptake stays in the upper ocean

Line 474; “total emitted MP inventory” -- How much is the total MP inventory?

Line 472; “total ocean MP inventory (second row, Fig. S1) and maximum surface concentration in

gyres” -- How much is the maximum surface concentration in gyres?

Line 480; “total passive inventory” -- How much is total passive inventory?

Latitudinal MP distributions are sensitive to the fraction assigned a rise rate

Line 487; “applied buoyancy assumptions” -- How then is it possible to have accurate results if they are assumptions? What are those assumptions?

thredds.geomar.de/thredds/catalog/open_access/kvale_et_al_2020_npg/catalog.html -- Appears that message “HTTP Status 404 - Not Found; THREDDS Data Server Version 4.6 -- Documentation”

References

8. Day, R. H. & Shaw, D. G. Patterns in the abundance of pelagic plastic and tar in the north pacific ocean, 1976–1985. *Mar.Pollut. Bull.* 18, 311 – 316, DOI: [https://doi.org/10.1016/S0025-326X\(87\)80017-6](https://doi.org/10.1016/S0025-326X(87)80017-6) (1987). -- The DOI does not work.

Lindeque, P. K. et al. Are we underestimating microplastic abundance in the marine environment? a comparison of 355 microplastic capture with nets of different mesh-size. *Environ. Pollut.* 265, 114721, DOI: <https://doi.org/10.1016/j.envpol.2020.114721> (2020). -- The DOI does not work.

Wichmann, D., Delandmeter, 359 P. & van Sebille, E. Influence of near-surface currents on the global dispersal of marine 360 microplastic. *J. Geophys. Res. Ocean.* 124, 6086–6096, DOI: 10.1029/2019JC015328 (2019). <https://agupubs.onlinelibrary.wiley.com/doi/pdf/10.1029/2019JC015328>. -- The link does not work. When you click it, it does not take the whole link as one line.

Zhu, L., Zhao, S., Bittar, T. B., Stubbins, A. & Li, D. Photochemical dissolution of buoyant microplastics to dissolved 391 organic carbon: Rates and microbial impacts. *J. Hazard. Mater.* 383, 121065, DOI: <https://doi.org/10.1016/j.jhazmat.2019.392> 121065 (2020). -- The DOI does not work.

Page 14 -- The references have a larger gap among them.

Points to be considered;

There is an excessive use of characterisations of “large”, “small”, “tiny”. You clarified the size of MP in your introduction (MP; 0.1 to 5 mm in diameter). These adjectives make the reader to think that the particles are larger or smaller the referred size of MP. If this is the case, there is the need to clarify the size.

Did you take into consideration the disadvantages of using UVic ESCM?

an absolute surface temperature that’s a bit too low

projections of ocean heat uptake that are a bit too high

In what level these disadvantages can influence the interpretation of the data?

We have included more caveats in the Discussion section, in line with the comments of the other reviewers.

Figures, the numbers are small and blurry. The figures should be in bigger size in order to be easily readable.

Finally, I would like to recommend to the authors the article below:

“Bioavailability and effects of microplastics on marine zooplankton”, Zara L.R. Botterell, Nicola Beaumont, Tarquin Dorrington, Michael Steinke, Richard C. Thompson, Penelope K.

Lindeque, <https://www.sciencedirect.com/science/article/pii/S0269749118333190>

Thank you for the reference, we have now included it in our manuscript.

Reviewer #3 (Remarks to the Author):

Review of Kvale et al. "Zooplankton grazing of microplastic can accelerate global loss of ocean oxygen"

This study describes an interesting new potential pathway of how oxygen concentration in the ocean could be affected by microplastics, a hitherto unknown process. I like the idea – provocative and important. Who knows whether it is correct, but it is logical and in my view deserves more attention in the future. However, there are a number of comments that the authors should address, mainly around biological realism and better articulating the model limitation and explaining things, which will make the study stronger:

The authors would like to thank Prof. Richardson for his careful consideration of our manuscript, which has helped us to improve its quality. Please find changes to the text in red font.

1. Note that as you define microplastics (down to 0.1 mm), zooplankton eat very few particles in this size range. Most of the phytoplankton zooplankton eat are much smaller than this. Further, not many NPZD models or ESMs model phytoplankton >100 um in size (virtually none?). Maybe you need to be talking about mini-microplastics down to 5 um in size (Brandon et al. 2019)?

We apologize and are grateful for pointing out the typo in the definition of microplastic, which was supposed to say 0.1 um rather than 0.1 mm, as used, e.g., by Galloway et al., 2017. We are aware that feeding interactions, like most other biological processes in the plankton, depend on size. The representation of these size-related plankton processes is highly simplified in our earth system model, given the simplistic representation of the plankton community. We therefore never explicitly simulate size and consider all plastic in the model as "biologically active". In other words, we only model the biologically relevant size fraction. Some plankton parameters, like grazing preference, implicitly consider size effects). We have now included mention of particle size in the Discussion (Line 133, 144) and Methods (Line 190, 205, 214).

2. You say in the Abstract "Although significant uncertainty accompanies these estimates,". Fig. 4 suggest that you have very little uncertainty in your model. I suspect there is no structural uncertainty in your ESM wrt phytoplankton or zooplankton functional groups. It is not clear even how many functional groups of zooplankton are in the model – is there one or two? I'd like to see some discussion of the caveats of the limited structural uncertainty in the model. I understand that ESMs have very basic zooplankton (and phytoplankton), but your results could be quite different depending on the zooplankton group (e.g. copepods, larvaceans, salps, krill or jellyfish – herbivores or carnivores) that were present. This should all be in a caveats section

This is an excellent point, and we fully agree with the reviewer that the ESM with only one zooplankton compartment is extremely simplistic when it comes the representation of the planktonic food web. However, this simplicity has been necessary to produce long-term spun-up simulations, and our results provide strong motivation to explore food web mediated MP effects in more complex models.

In our model, the grazing preference implicitly considers the overall "bioavailability" of the microplastic to generic zooplankton consumption. Adding complexity in the micro, meso- and macrozooplankton fractions (e.g., enhanced near-surface recycling, deep vertical migration) would shift pathways between microbial-loop recycling and deeper export. Furthermore, adding flexibility in feeding responses, for example by using acclimative feeding (e.g., Holling Type 3 implying varying instead of fixed grazing preferences) would make distributions and pathway regionally or seasonally variable. We have added more caveats regarding structural uncertainty to the Discussion section (Lines 133, 143) and expanded the Methods model description (from Line 164).

3. On the theme of what should be mentioned in a caveats section, I'd like to see more discussion of this. You are focusing on export production and oxygen concentration – what about diel vertical

migration, different sinking rates of different zooplankton functional groups, marine snow formation by larvaceans? All of these can affect export production and oxygen concentration in deeper layers, but I suspect none are in the model.

It is correct that our model contains a very simplistic (although, state-of-the-art for an earth system model) representation of ecological interactions. For example, vertical migration of zooplankton is still the exception in global-scale models and has only started to become more prominent in the last few years (e.g., Bianchi et al. 2013, Aumont et al. 2018). Greater complexity often fails to improve model performance against observations of nutrients, carbon, and oxygen and tuning food webs is challenging because of the lack of zooplankton data for groups other than maybe copepods on the global scale. We know there are a lot of important processes not explicitly represented, but complex models involve a large degree of parameter uncertainty, are difficult to interpret and do not guarantee improved degrees of realism (e.g., Kriest, 2017). A notable effect in a simple model generally motivates future research using more complex models. This probably also applies to plastics modelling. A sentence is added to the Discussion (Line 145).

4. I'd like to see some justification for why you have included marine snow? Zooplankton eat microplastics if they are not part of marine snow. And only some zooplankton eat marine snow, whilst others eat phytoplankton. More justification is needed about the model structure

In our model, zooplankton do not eat plastic contained in marine snow (to keep things simple in this first model version). Marine snow aggregation of microplastic is included in our simulations because it provides a sink for the "free" microplastic particles, removing it from availability to the zooplankton. The Methods section now includes this information (Line 187).

5. The main reference you use for convincing the reader that zooplankton ingest microplastics is Cole et al. (2013). Can you cite some more recent work as well that confirms that zooplankton eat microplastics?

The Botterell et al. (2019) review is now cited as well.

6. Little evidence is provided about the selectivity of microplastics by zooplankton. In the study you run a range of simulations from low to high ingestion, but I'd like to see some references showing that this is what happens in zooplankton. For example, is there evidence of zooplankton preferring microplastics to phytoplankton/marine snow (i.e. your TestLow scenario)

The Botterell et al. (2019) review lists a variety of ingestion responses regarding different types of plastic for different species. How to summarise this information in a globally-appropriate single parameter value is unknown, which is why we test a range of assumptions. This is now discussed in greater detail in the Discussion and Methods sections (from Lines 129, 202).

7. One piece of evidence you might like to include explaining why zooplankton ingest microplastics is that a biological layer colonises the particle, so that zooplankton think it is biological in nature (e.g. Dudek et al. 2020 *Limnology and Oceanography Letters*)

Thank you for pointing out this paper. This reference has been added to the Methods model description.

8. I think your message would be more effective if you were not talking in model speak with some of your terms. It is not reader-friendly using terms such as TestLo, TestHi, TestMed, NoGraz. This is throughout the figures too. Please make it easy on the reader and use more descriptive terms. You can make it longer to make it easier

You are absolutely right. Simulations have been renamed to "No Bio", "Low Concentration, High Preference", "High Concentration, Low Preference", and "Moderate Concentration, Balanced Preference".

9. You should be consistent in describing the zooplankton feeding preference. Sometimes you say

“zooplankton feeding preference for marine snow” and other times you say “zooplankton feeding preference for microplastics”. This is likely to confuse readers. Please be consistent
Each simulation has unique grazing preferences both for microplastics and for each of the organic food sources. Grazing preference is now better explained in the Methods section (Line 202).

10. Fig. 3. I like the idea but think it can be executed better. Have a key explaining the symbols. The typesetting for A. and D. are poor and it seems like the export is going into them rather than the deep ocean?

Thanks for the suggestion. We have revised Figure 3 according to the journal’s formatting requirements with the help of a graphic designer.

11. A logical step in your explanation is missed in the text and in Fig. 3. With zooplankton ingesting microplastics in nutrient replete system, reduces grazing pressure on phytoplankton and then leads to more export production, misses the step of phytoplankton increase in between. More explanation is needed to help the reader. This is especially true because the phytoplankton don’t really increase in nutrient-deficient areas because they rely on recycled nutrients. It will help the reader understand if you provide more description of the process.

We purposefully did not focus the discussion on biomass and primary production trends because biomass trends are not informative of export production trends (these two factors are frequently decoupled due to temperature effects). We have revised our explanations in the text of the mechanisms (Lines 55, 59) and we have also revised the caption of Fig. 3.

12. I like the idea of Fig. 3, but feel it could be improved and made clearer. For example, I don’t find the reduction in the recycling by zooplankton is included in the Not nutrient limited systems. Would a sequence of panels for each of the Nutrient limited and Not nutrient limited systems better describe the changes in the system that happen?

We have revised Figure 3 according to the journal’s formatting requirements with the help of a graphic designer. We do not emphasise a reduction in recycling by zooplankton in the non-nutrient-limited system because this is not a dominating (or even always observed) effect. What matters in the nutrient limited regions is the change in relative proportions between the microbial loop recycling and zooplankton excretion, but in the nutrient replete system this shift is not a useful indicator of system response.

13. You say “At a global scale, the additional loss of oxygen in the regions not limited by nutrients exceeds the gain of oxygen in regions limited by nutrients.” It is very difficult to see the relative size of different areas in your plots because it appears you have not used a projection. Please use a projection in your figures that better represents the area of each ocean

Thanks for this suggestion. The figures have been changed.

14. In the regions that are nutrient replete and show an increase in primary production, will there be an increase in carbon dioxide drawdown from the ocean to the atmosphere? i.e. Does your work suggest the biological pump be stronger? Please comment

Our simulations show a 0-200+% increase in air-sea flux of carbon between 40-60 S at year 2020 in the Southern Ocean. However, the Eastern Tropical Pacific shows no change in carbon flux and the effects in the North Pacific are mixed. In the tropics the effect is also mixed, with both strong increases and strong decreases in regional air-sea fluxes that can exceed +-200% of the control simulation rates. Linking plastic-induced modification of carbon flux to the biological carbon pump is a complex topic we prefer to leave to subsequent publications. Partly this is due to the problem of timescales and the interior-ocean transport and resurfacing of carbon anomalies on timescales of decades and similar to those of changes in plastic input. This is now discussed at the end of the Discussion section (Line 148).

15. Figure 4 needs axis labels

The figure has been changed.

16. eds to be more description of the model – how many phytoplankton groups, how many zooplankton groups. There is no reason why at least the very basics of the biogeochemical part of the ESM could not be explained.

The model description has been expanded in the Methods section.

17. I'd like to see some suggestions of where to go next with this work? What are the next steps?

A couple of sentences have been added to the end of the Discussion (Line 152).

18. Figure 4 needs axis labels

The figure has been changed.

19. Fig. 2. I don't understand. What does No Graz mean everywhere? Please explain

No Graz is the control simulation that does not include grazing of microplastic by zooplankton (or aggregation of microplastic in marine snow), as explained in the text (Line 30). It has been renamed "No Bio".

20. There are relatively few references in this study, but that might be a limitation of the journal. I found quite a few statements needed support by reference and didn't have it.

We have included more references in the revised version.

Anthony J. Richardson

References:

Aumont et al. 2018, doi: 10.1029/2018GB005886

Bianchi et al. 2013, doi: 10.1038/NGEO1837

Reviewer #4 (Remarks to the Author):

Kvale et al present an ambitious modelling-centric paper that looks to link laboratory derived data on zooplankton to global oxygen loss in response to microplastics. Unfortunately, the paper has made too many leaps, statements are vague, many steps have incorrectly selected/understood data from other sources, and lacks transparent methodology. I cannot recommend this paper for publication. Additional comments below:

The authors would like to thank the reviewer for carefully reading our manuscript and for providing comments that we have used to improve its clarity. Several of the comments were similar to those made by the other reviewers, and have already been corrected. Others are based on a mis-understanding of our work, which we have tried to avoid for the revised version by a more comprehensive description of the approach, the assumptions and the technical backgrounds. Changes in the manuscript are made in green font.

Title - misleading, as in Fig 2E the authors show oxygen loss and decline depending on latitude

This is an incorrect interpretation of the figure. Naturally, our figures show a correlation with latitude as oceanic provinces, like oligotrophic vs. nutrient-replete regimes, roughly correlate with latitude. However, latitude is not a driver of oxygen loss and we stand by our title. It is possible the reviewer mis-interprets Figure 2E (first panel on the left, now Figure 5) which shows generally increasing water column O₂ inventory with latitude, which is to be expected for reasons of solubility. The other panels of Figure 2E demonstrate an *additional* loss of oxygen in response to the addition of microplastic, which broadly follows latitude but not exactly (see the Eastern Tropical Pacific, a nutrient-replete ecosystem).

L15 - a lower bound of 0.1 mm = 100 micron which is above size range consumed by many zooplankton; lower range would be 0.001 mm.

Thank you for pointing out this typo, the sentence should have stated 0.1 μm (not mm), e.g., as used in Galloway et al. 2017.

Please also see our response to the respective comment from Reviewer 3.

L19 and elsewhere - Throughout the paper there is heavy reference to reference 6, coined as a companion MS. This has not been peer-reviewed as yet, and this reviewer has no way to know whether the methodology used is appropriate.

Our Scientific Reports manuscript is now published and freely available at <https://www.nature.com/articles/s41598-020-72898-4>.

"significant influence of marine snow and zooplankton fecal pellets" -- this is a big statement that cannot be checked owing to reference 6 not being available; but there are no caveats given either, is this based on experimental data only? such experiments often have caveats of very high MP numbers, single type of MP etc. these caveats and errors are not carried forward sufficiently.

Our Scientific Reports manuscript is now published and freely available. This statement is based upon that study (and cites it), which contains an extensive search of the parameter space as well as discussion of assumptions and uncertainties. Also, the word before "significant" is "potentially", which changes the strength of the statement in question. We prefer to limit discussion of the transport effects in this manuscript, as the focus here is on biogeochemistry and the transport is discussed in detail in the other article.

Test Lo/Med/Hi are referred to in text, but these are arbitrary. There is no data or indication of the extent of the low/high settings and whether they're in anyway realistic. Fig 1 legend doesn't state what these codes properly refer to (i.e. marine snow influence).

Yes, these definitions are arbitrary in the sense that they are "highlighted to represent the solution space of the 14 individuals of a 300 member Latin Hypercube parameter search that produced plausible global microplastic inventories and subsurface particle maxima, using pollution rates and marine snow aggregation rates within available estimates." (as stated in

the Results, when the three simulations are introduced (Line 30)). We have renamed the simulations for better clarity. Parameter settings for the three simulations are given in the Scientific Reports article and we now refer to it in Line 174. Another sentence is added mentioning realism (Line 44). The figure labelling has been improved.

Fig 1 text in axis is too small to read.

The figure labelling has been improved.

L39 "Microplastic offers zooplankton an alternative food source" - no, from what I have read it is captured alongside their food.

Thank you for pointing out this confusing sentence. Microplastic is an alternative food source in the sense that if it was not present, it would not be eaten. Since it is there, the model zooplankton have the opportunity to consume it. A globally-appropriate parameterisation of grazing selection of microplastic as opposed to other food sources for a single zooplankton type is highly uncertain, which is why we present three models utilising several grazing preferences for microplastic. Grazing preference is now better explained in the manuscript (Lines 36, 40, 51, 133, 203).

L57 - "depending on model configuration" - be specific.

We apologize for the unclear wording. We have rephrased the sentence.

L62 - zooplankton grazing preference for MP? I have seen no evidence that they would selectively graze on MP over algae, quite the opposite.

The "zooplankton grazing preference" is the name of a parameter used in our biogeochemical model that seems to produce confusion among non-modellers (or those who work in plastics). It represents prey selection, and in our simulations it can represent a relative dislike of plastic compared to other foods. In TestHi (now called "low concentration, high preference", zooplankton avoid MP ingestion if other foods are available. In TestMed (now called moderate concentration, balanced preference"), zooplankton treat MP roughly equal to other foods. In TestLo (now called "low concentration, high preference"), zooplankton prefer grazing on MP compared to other food sources. How to parameterise prey selection for a single zooplankton type in a global earth system model, against a single microplastic type, is very uncertain (which is why we explore 3 possible preferences). Uncertainty is now mentioned with respect to grazing preference as well as size classes in the Discussion (Line 129), based also on comments of Reviewers 1&3.

Methods - massive lack of transparency here, with no detail given.

The Methods section has now been substantially expanded, also incorporating specific comments from the other reviewers.

Better justification of microplastic loadings are required; why start at 1950 for "micro"plastic? How were the microplastic levels in the model comparing with environmental levels actually observed. This is never stated in text and I don't feel the MP loads are elucidated in the figures. How much plastic is causing the observed shifts in the models?

Plastic emissions are taken from primary literature, cited in the Methods section (Geyer et al., 2017, Jambeck et al., 2015), and a range of input fractions as "microplastic" were extensively tested in our Scientific Reports paper (<https://www.nature.com/articles/s41598-020-72898-4>), which examined the question of transport and regional accumulation. We show here only a sub-section of the simulations produced in that other paper, which have been filtered based on selection criteria laid out in the Methods section. The historical record of microplastic ocean observations is too sparse to be meaningful for global model assessment. Figure 1 is the best approximation we can produce from our modelling framework of what Eriksen et al., 2014 simulated in their extensive compilation of c.a. year

2010 surface data. This is the most comparable estimate of “load” we can make, since our Eulerian methodology is so different from previous efforts. We have added a sentence about realism in the Results based on an earlier comment. As pointed out in response to a comment from Reviewer 1, the second major result of our paper is that “load” is secondary to grazing preference in determining the biogeochemical impact of pollution, and the text has been expanded to enhance this finding (Line 131).

L127 - "40% of total plastic production enters the oceans"... no, this is factually incorrect. The Jambeck paper consider a hypothetical extreme of 40% of mismanaged waste from coastal populations 50km from coasts might end up in ocean, but this is not the same as you state. Thank you for pointing out this typo. We have now corrected it.

Kriest, I.: Calibration of a simple and a complex model of global marine biogeochemistry, *Biogeosciences*, 14, 4965–4984, <https://doi.org/10.5194/bg-14-4965-2017>, 2017.

REVIEWER COMMENTS

Reviewer #3 (Remarks to the Author):

I think the authors have addressed my comments well, with only relatively minor differences in opinion.

Anthony J. Richardson

Reviewer #5 (Remarks to the Author):

Review of "Zooplankton grazing of microplastic can accelerate global loss of ocean oxygen" by Kvale et al.

Using a numerical model incorporating biogeochemical and microplastic processes in the world's oceans, the authors demonstrated that grazing pressure by zooplankton to phytoplankton decreases owing to ingestion of microplastics, and that dissolved oxygen in the ocean is likely to decrease due to the enhanced decomposition of detritus increased by redundant phytoplankton.

The above scenario provided by the authors is indeed interesting, and so the present study potentially plays a role to advance earth system sciences as well as marine plastic pollution research. I however have criticisms itemized below at the present time. It is much appreciated if the authors would add revisions or rebuttal to erase my concerns; otherwise withdrawing the manuscript is an option.

1. The authors should reasonably justify the range of the grazing preference parameter ($0.132 < \phi_{MP} < 0.260$), because the conclusions in the present study depend on this parameter range. To the best of my knowledge, there is no study stating that zooplankton prefer microplastics. The grazing of microplastics by zooplankton is not due to an active choice by zooplankton, but due to an accidental ingestion along with phytoplankton. Thereby, it seems likely that the grazing rate of microplastics by zooplankton depends only on the ratio of abundance between microplastics and phytoplankton. The previous studies have reported that the abundance of pelagic microplastics is < 1.0 pieces/m³ in the actual open oceans, and that phytoplankton is much more abundant. Therefore, the abundance ratio (preference parameter in the present study) between 0.1 and 0.3 seems to be too much.

2. Overall, the validation of the model accuracy was quite insufficient in the manuscript. Numerical model approach always requires a careful and quantitative validation using actually observed data; otherwise accomplishments by the numerical model approach remains less convincing. In particular, the present model is very complex because of incorporating a variety of compartments such as nutrients, phyto- and zooplankton, detritus, and microplastics in different phase. Nonetheless, the model validation was conducted roughly by a single sentence (lines 32-34: These three simulations...within available estimate"). The quantitative validation with observed data was not conducted in their comparison manuscript published in Scientific Reports. I recognize that the microplastic abundance might be difficult to compare with the actual data, because field surveys of oceanic microplastics were still sparse at the present time. However, archived and satellite datasets were available for comparison with nutrients and plankton. A subsection at least should be added for the model validation.

3. I questioned a large gap in the sizes of modeled microplastics. The authors compared modeled abundance of microplastics with values in Eriksen et al. (2014) (lines 32-34: These three simulations...within available estimate"), who collected plastic fragments using towing nets with mesh sizes larger than 0.3 mm. This means that the sizes of microplastics reproduced in the present model were also larger than 0.3 mm. However, microplastics ingested by zooplankton were much smaller than these microplastics (0.1~0.3 mm; e.g., table 2 in Desforges et al., 2015, Arch Environ Contam

Toxicol). Explicitly, the microplastics reproduced in the present model should not be ingested by zooplankton in the reality.

Specific points:

4. Lines 40-46, "The Low Concentration...Low Preference simulation": I was confusing the names for these simulation cases. I recognized that the parameter tuned by the authors was the grazing preference parameter only. So, it was enough to refer the three simulations as "Low Preference", "High Preference", and "Balanced Preference". The name "Low Concentrations, High Preference" sounds that two parameters were tuned for the model setup: one was for the concentration, while the other was for preference.

5. "Southern Ocean response and global trend": It was surprising that the oxygen loss was remarkable in the Southern Ocean (Fig. 5) despite the relatively small abundance of microplastics in the ocean (Fig. 1). It seems likely that the authors provided no interpretation for this discrepancy in the manuscript.

6. Line 129-132, "This suggests both...surface microplastic accumulation": Was this argument for the discrepancy found in the Southern Ocean (my comment 5)? If this is true, I have the same comment as comment 5. Please provide the reasonable explanation for the discrepancy: large oxygen depression in spite of small microplastic abundance.

7. Figure 2, "dimensionless": Why did the flux (e.g., transport per unit time) normalized by biomass (e.g., weight per unit seawater volume) become a dimensionless number?

Reviewer #5 (Remarks to the Author):

Review of “Zooplankton grazing of microplastic can accelerate global loss of ocean oxygen” by Kvale et al.

Using a numerical model incorporating biogeochemical and microplastic processes in the world's oceans, the authors demonstrated that grazing pressure by zooplankton to phytoplankton decreases owing to ingestion of microplastics, and that dissolved oxygen in the ocean is likely to decrease due to the enhanced decomposition of detritus increased by redundant phytoplankton.

The above scenario provided by the authors is indeed interesting, and so the present study potentially plays a role to advance earth system sciences as well as marine plastic pollution research. I however have criticisms itemized below at the present time. It is much appreciated if the authors would add revisions or rebuttal to erase my concerns; otherwise withdrawing the manuscript is an option.

The authors would like to thank the Reviewer for their thoughtful suggestions for our manuscript. Author responses are found below. Changes to the manuscript can be found in orange font in the accompanying text.

1. The authors should reasonably justify the range of the grazing preference parameter ($0.132 < \text{phiMP} < 0.260$), because the conclusions in the present study depend on this parameter range. To the best of my knowledge, there is no study stating that zooplankton prefer microplastics. The grazing of microplastics by zooplankton is not due to an active choice by zooplankton, but due to an accidental ingestion along with phytoplankton. Thereby, it seems likely that the grazing rate of microplastics by zooplankton depends only on the ratio of abundance between microplastics and phytoplankton. The previous studies have reported that the abundance of pelagic microplastics is < 1.0 pieces/m³ in the actual open oceans, and that phytoplankton is much more abundant. Therefore, the abundance ratio (preference parameter in the present study) between 0.1 and 0.3 seems to be too much.

We agree the ratio of the abundance of microplastic to organic food sources is probably very important to determining ingestion rates. This information is included in the calculation of microplastic ingestion. We have added the ingestion equation to the Methods section and expanded our description of it (Lines 218-236). We have also clarified, throughout the text, that the grazing preference is actually a relative grazing selectivity (not an abundance ratio, as suggested). This relative grazing selectivity includes more than just a straight-forward demonstrated preference or aversion for a plastic particle by a zooplankton (which might be measured in a lab), but implicitly represents biofouling (prey confusion), accidental ingestion, or size selectivity for all zooplankton represented by our single zooplankton functional type. To our knowledge, this is a necessary parameter in terms of mathematical formulation that cannot be constrained at the global scale because it integrates a multitude of processes down to organism scale. This is why we tested a range of values from weaker to stronger selectivity in 300 simulations, and show model results from all 14 simulations that fit our Latin Hypercube criteria in Fig.8.

What we found with parameter sensitivity testing is that even when microplastic particle concentrations are low relative to other food sources, there could still be a lot of uptake because of the strong sensitivity of the model to the grazing selectivity. We acknowledge and discuss the high degree of uncertainty in this parameter, which is why we draw only qualitative conclusions with our study (Line 147) and urgently recommend more research into this specific aspect (Line 140). One of the significant findings of this study is that even when zooplankton are assigned a weak selectivity for microplastic as a food source and surface concentrations are fairly low, biogeochemical impacts are still possible. We have tried to further clarify the discussion of this parameter in the manuscript (Line 144, and lines listed above).

2. Overall, the validation of the model accuracy was quite insufficient in the manuscript. Numerical model approach always requires a careful and quantitative validation using actually observed data; otherwise accomplishments by the numerical model approach remains less convincing. In particular, the present model is very complex because of incorporating a variety of compartments such as nutrients, phyto- and zooplankton, detritus, and microplastics in different phase. Nonetheless, the

model validation was conducted roughly by a single sentence (lines 32-34: These three simulations...within available estimate”). The quantitative validation with observed data was not conducted in their comparison manuscript published in Scientific Reports. I recognize that the microplastic abundance might be difficult to compare with the actual data, because field surveys of oceanic microplastics were still sparse at the present time. However, archived and satellite datasets were available for comparison with nutrients and plankton. A subsection at least should be added for the model validation.

We fully agree that model “assessment” is necessary and useful. The biogeochemical model is unchanged from previously published versions, which is why we cite the thorough and quantitative model-data assessment of Kvale et al., 2015, as well as previous studies using the model, in the Methods section (as is standard-practice in our field). Interested readers should examine Kvale et al., 2015 in particular, as it offers all the details of the biogeochemical model equations and assumptions, and an extensive comparison to available phytoplankton and nutrient data.

We disagree that the microplastic model is assessed “roughly by a single sentence”- this comment might arise due to a misunderstanding of the Latin Hypercube methodology. We have expanded our description of this method in the manuscript (Lines 178-181). This model is not tuned to observations, which are too sparse for microplastic distribution assessment at the global scale. We cite the Eriksen et al. (2014) data-constrained modelling effort and note its similarity to the MC simulation (Line 43), but comparisons to this (or any) available microplastic survey are problematic for the reasons given by the Reviewer at point 3 below. However, we have searched the literature for recent observational references and have added a few new ones (Lines 45-49, 186).

The Latin Hypercube presents an alternative approach to model tuning, in which we explore the entire parameter space and then attempt to eliminate certain combinations based on what observations are available. We present this entire parameter space in the Scientific Reports manuscript- 700 simulations in total. Because of the very large uncertainties in this model, which is the first global model of microplastic/biological interaction, “validation” would be inappropriate. It cannot be validated, but falsified for certain parameter ranges that can be excluded, leaving a plausible parameter range for which the model results are described. The Supplement of the Scientific Reports manuscript provides an extensive parameter sensitivity assessment of the model. In this manuscript, we choose to describe 3 of the 14 simulations that met a set of criteria (Line 180-188):

- 1) A plausible microplastic global inventory (based on a cited study)
- 2) A plausible global pollution rate (based on a cited study)
- 3) A plausible marine snow aggregation fraction (based on a cited study)

The remaining 11 simulations are not described in the manuscript for reasons of length and interest, although they are included in the global oxygen loss figure to show the whole ensemble sensitivity.

We acknowledge the large uncertainty in our results due to the under-constrained parameters- see the large range in potential near-surface microplastic inventories in Figure 1! The model results are within this uncertainty range that is consistent with the observations. It is precisely because of the large uncertainties in this model that we are very careful in what we conclude in our manuscript, and hope the specific points we raise can motivate future research directions.

3. I questioned a large gap in the sizes of modeled microplastics. The authors compared modeled abundance of microplastics with values in Eriksen et al. (2014) (lines 32-34: These three simulations...within available estimate”), who collected plastic fragments using towing nets with mesh sizes larger than 0.3 mm. This means that the sizes of microplastics reproduced in the present model were also larger than 0.3 mm. However, microplastics ingested by zooplankton were much smaller than these microplastics (0.1~0.3 mm; e.g., table 2 in Desforges et al., 2015, Arch Environ Contam Toxicol). Explicitly, the microplastics reproduced in the present model should not be ingested by zooplankton in the reality.

The model does not simulate microplastic particle size. We have added another sentence to this effect in the Methods (Line 204). The Eriksen dataset was available so we considered it, but obviously it is an imperfect analogue to our simulated particles (one reason we keep 3 and 14 model simulations in the main analysis). We are not aware of a better dataset to compare our model to. The

sentence is modified to highlight the potential size bias (Line 44).

Specific points:

4. Lines 40-46, “The Low Concentration...Low Preference simulation”: I was confusing the names for these simulation cases. I recognized that the parameter tuned by the authors was the grazing preference parameter only. So, it was enough to refer the three simulations as “Low Preference”, “High Preference”, and “Balanced Preference”. The name “Low Concentrations, High Preference” sounds that two parameters were tuned for the model setup: one was for the concentration, while the other was for preference.

We apologise for the confusion. In fact, more parameters than just the relative grazing selectivity differ between simulations. These simulations were produced using Latin Hypercube methodology on the whole parameter space. This is now made more explicit (Line 33), with a table added (Table 1). The simulations have been renamed “High Concentration”, “Moderate Concentration”, and “Low Concentration”.

5. “Southern Ocean response and global trend”: It was surprising that the oxygen loss was remarkable in the Southern Ocean (Fig. 5) despite the relatively small abundance of microplastics in the ocean (Fig. 1). It seems likely that the authors provided no interpretation for this discrepancy in the manuscript.

Figure 3b&c illustrates the mechanism of oxygen loss in the Southern Ocean. Another sentence is added to the section to state this more explicitly (Line 113).

6. Line 129-132, “This suggests both...surface microplastic accumulation”: Was this argument for the discrepancy found in the Southern Ocean (my comment 5)? If this is true, I have the same comment as comment 5. Please provide the reasonable explanation for the discrepancy: large oxygen depression in spite of small microplastic abundance.

The new sentence “This impact occurs because the region is not macronutrient-limited, therefore even slight alleviation of grazing pressure increases export production” added in response to the above point explicitly states the reason for strong deoxygenation despite low microplastic concentrations (Line 113). Please also notice that we have added a new paragraph to the Discussion (Lines 130-138) to discuss possible impacts of iron limitation on our results.

7. Figure 2, “dimensionless”: Why did the flux (e.g., transport per unit time) normalized by biomass (e.g., weight per unit seawater volume) become a dimensionless number?

Thank you for pointing out this typo. The unit is per year and this has been corrected.

The authors would again like to thank the Reviewer for their efforts.

REVIEWERS' COMMENTS

Reviewer #5 (Remarks to the Author):

I recognized that the authors sincerely addressed my previous concerns. I am now delighted to recommend this manuscript to be published in your journal.